# An Embarrassingly Simple Approach to Semi-Supervised Few-Shot Learning

**Xiu-Shen Wei**[1,2], **He-Yang Xu**[1], **Faen Zhang**[3], **Yuxin Peng**[4,*] **Wei Zhou**[5]

[1]School of Computer Science and Engineering, Nanjing University of Science and Technology
[2]State Key Laboratory of Integrated Services Networks, Xidian University
[3]Qingdao AInnovation Technology Group Co., Ltd
[4]Wangxuan Institute of Computer Technology, Peking University
[5]CICC Alpha (Beijing) Private Equity

## Abstract

Semi-supervised few-shot learning consists in training a classifier to adapt to new tasks with limited labeled data and a fixed quantity of unlabeled data. Many sophisticated methods have been developed to address the challenges this problem comprises. In this paper, we propose a simple but quite effective approach to predict accurate *negative* pseudo-labels of unlabeled data from an indirect learning perspective, and then augment the extremely label-constrained support set in few-shot classification tasks. Our approach can be implemented in just few lines of code by only using off-the-shelf operations, yet it is able to outperform state-of-the-art methods on four benchmark datasets.

## 1 Introduction

Deep learning [16] allows computational models that are composed of multiple processing layers to learn representations of data with multiple levels of abstraction, which has already demonstrated its powerful capabilities in many computer vision tasks, *e.g.*, object recognition [7], fine-grained classification [39], object detection [18], etc. However, deep learning based models always require large amounts of supervised data for good generalization performance. Few-Shot Learning (FSL) [37], as an important technique to alleviate label dependence, has received great attention in recent years. It has formed several learning paradigms including metric-based methods [29, 33, 45], optimization-based methods [4, 25, 28], and transfer-learning based methods [3, 24].

More recently, it is intriguing to observe that there has been extensive research in FSL on exploring how to utilize unlabeled data to improve model performance under few-shot supervisions, which is Semi-Supervised Few-Shot Learning (SSFSL) [9, 15, 19, 23, 36, 44]. The most popular fashion of SSFSL is to predict unlabeled data with pseudo-labels by carefully devising tailored strategies, and then augment the extremely small support set of labeled data in few-shot classification, *e.g.*, [9, 15, 36]. In this paper, we follow this fashion and propose a simple but quite effective approach to SSFSL, *i.e.*, a Method of sUccesSIve exClusions (MUSIC), cf. Figure 1.

As you can imagine, in such label-constrained tasks, *e.g.*, 1-shot classification, it would be difficult to learn a good classifier, and thus cannot obtain sufficiently accurate pseudo-labels. Therefore, we

*Corresponding author. X.-S. Wei and H.-Y. Xu are with Key Lab of Intelligent Perception and Systems for High-Dimensional Information of Ministry of Education, and Jiangsu Key Lab of Image and Video Understanding for Social Security, Nanjing University of Science and Technology. This work was supported by National Key R&D Program of China (2021YFA1001100), National Natural Science Foundation of China under Grant (62272231, 61925201, 62132001, U21B2025), Natural Science Foundation of Jiangsu Province of China under Grant (BK20210340), the Fundamental Research Funds for the Central Universities (30920041111, NJ2022028), CAAI-Huawei MindSpore Open Fund, and Beijing Academy of Artificial Intelligence.

36th Conference on Neural Information Processing Systems (NeurIPS 2022).

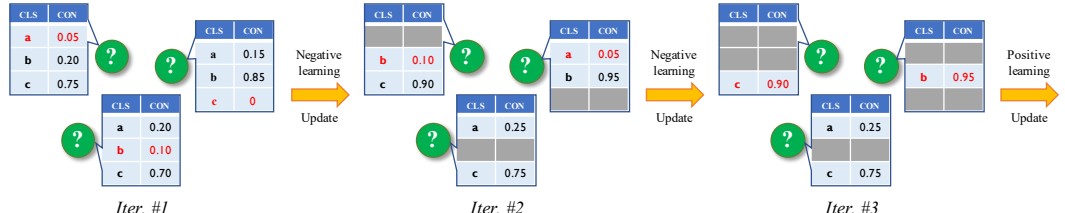

Figure 1: Pipeline of our MUSIC approach, where we take 3-way few-shot classification for an example. Specifically, the green data points marked with question mark are unlabeld data in SSFSL, "CLS" represents the classes of data, and "CON" stands for the predicted confidence of the corresponding pseudo-labels. In each iteration, we highlight the selected (negative/positive) pseudo-label with the red color, and exclude that class in the following iteration. Note that, the unlabeled data at the bottom is not returned with the final positive label due to our reject option strategy ($\delta = 0.2$), cf. Eqn. (4).

think about the problem in turn, and realize the process of pseudo-labeling in SSFSL as a series of successive exclusion operations. In concretely, since it is hard to annotate which class the unlabeled data belongs to, in turn, it should be relatively easy[2] to predict which class *it does not belong to* based on the lowest confidence prediction score. Thus, if we treat the predicted pseudo-labels in the previous traditional way as labeling positive labels, our exclusion operation is to assign *negative* pseudo-labels to unlabeled data. In the following, we can use the negative learning paradigm [10] to update the classifier parameters and continue the negative pseudo-labeling process by excluding the predicted negative label in the previous iteration, until all negative pseudo-labels are obtained. Moreover, it is apparent to find that when all negative labels of unlabeled data are sequentially excluded and labeled, their positive pseudo-labels are also obtained. We can thus eventually augment the small support set with positive pseudo-labels, and fully utilize the auxiliary information from both labeled base-class data and unlabeled novel-class data in SSFSL. Also, in our MUSIC, to further improve few-shot classification accuracy, we equip a minimum-entropy loss into our successive exclusion operations for enhancing the predicted confidence of both positive and negative labels.

In summary, the main contributions of this work are as follows:

- We propose a simple but effective approach, *i.e.*, MUSIC, to deal with semi-supervised few-shot classification tasks. To our best knowledge, MUSIC is the first approach to leverage negative learning as a straightforward way to provide pseudo-labels with as much confidence as possible in such extremely label-constrained scenarios.

- We can implement the proposed approach using only off-the-shelf deep learning computational operations, and it can be implemented in just few lines of code. Besides, we also provide the default value recommendations of hyper-parameters in our MUSIC, and further validate its strong practicality and generalization ability via various SSFSL tasks.

- We conduct comprehensive experiments on four few-shot benchmark datasets, *i.e.*, *miniImageNet*, *tieredImageNet*, *CIFAR-FS* and *CUB*, for demonstrating our superiority over state-of-the-art FSL and SSFSL methods. Moreover, a series of ablation studies and discussions are performed to explore working mechanism of each component in our approach.

## 2 Related Work

**Few-shot learning** The research of few-shot learning [4, 29, 33, 42, 45] aims to explore the possibility of endowing learning systems the ability of rapid learning for novel categories from a few examples. In the literature, few-shot learning methods can be roughly separated into two groups: 1) Meta-learning based methods and 2) Transfer-learning based methods.

Regarding meta-learning based methods, aka "learning-to-learn", there are two popular learning paradigms, *i.e.*, metric-based methods [29, 33, 45] and optimization-based methods [4, 25, 28]. More specifically, Prototypical Networks [29] as a classical metric-based method was considered

---

[2]Because the probability of selecting a class that does not belong to the correct label is high, the risk of providing incorrect information in doing so is low, especially for SSFSL.

to generate an embedding in which data points cluster around a single prototype representation for each class. DeepEMD [45] proposed to adopt the Earth Mover's Distance as a metric to compute a structural distance between dense image representations to determine image relevance for few-shot learning. For optimization-based methods, MAML [4] learned an optimization method to follow the fast gradient direction to rapidly learn the classifier for novel classes. In [25], it reformulated the parameter update into an LSTM and achieved this via a meta-learner.

Regarding transfer-learning based methods, they are expected to leverage techniques to pre-train a model on the large amount of data from the base classes, without using the episode training strategy. The pre-trained model is then utilized to recognize novel classes of few-shot classification. In concretely, [24] proposed to directly set the final layer weights from novel training examples during few-shot learning as a weight imprinting process. In [3], the authors investigated and shown such transfer-learning based methods can achieve competitive performance as meta-learning methods.

**Semi-supervised few-shot learning**    Semi-Supervised Learning (SSL) is an approach to machine learning that combines a small amount of labeled data with a large amount of unlabeled data during training [6, 46]. In the era of deep learning, SSL generally utilizes unlabeled data from the following perspectives, *e.g.*, considering consistency regularization [14], employing moving average strategy [30], applying adversarial perturbation regularization [22], etc.

In recent years, the use of unlabeled data to improve the accuracy of few-shot learning has received increasing attention [9, 15, 19, 23, 36, 44], which leads to the family of Semi-Supervised Few-Shot Learning (SSFSL) methods. However, directly applying SSL methods to few-shot supervised scenarios usually causes inferior results due to the extreme small number of labeled data, *e.g.*, 1-shot. More specifically, to deal with the challenging SSFSL, Ren *et al.* [26] extended Prototypical Networks [29] to use unlabeled samples when producing prototypes. TPN [19] was developed to propagate labels from labeled data to unlabeled data by learning a graph that exploits the manifold structure of the data. Recently, state-of-the-art SSFSL methods, *e.g.*, [9, 15, 36], were proposed to predict unlabeled data by pseudo-labeling and further augment the label-constrained support set in few-shot classification. Distant from previous work, to our best knowledge, we are the first to explore leveraging complementary labels (*i.e.*, negative learning) to pseudo-label unlabeled data in SSFSL.

**Negative learning**    As an indirect learning method for training CNNs, Negative Learning (NL) [10] was proposed as a novel learning paradigm w.r.t. typical supervised learning (aka Positive Learning, PL). More specifically, PL indicates that "input image belongs to this label", while NL means "input image does *not* belong to this complementary label". Compared to collecting ordinary labels in PL, it would be less laborious for collecting complementary labels in NL [10]. Therefore, NL can not only be easily combined with ordinary classification [5, 10], but also assist various vision applications, *e.g.*, [12] dealing with noisy labels by applying NL, [35] using unreliable pixels for semantic segmentation with NL, etc. In this paper, we attempt to leverage NL to augment the few-shot labeled set by predicting negative pseudo-labels from unlabeled data, and thus obtain more accurate pseudo labels to assist classifier modeling under label-constrained scenarios.

## 3    Methodology

### 3.1    Problem Formulation

**Definition**    In Semi-Supervised Few-Shot Learning (SSFSL), we have a large-scale dataset $\mathcal{D}_{base}$ containing many-shot labeled data from each base class in $\mathcal{C}_{base}$, and a small-scale dataset $\mathcal{D}_{novel}$ consisting of few-shot labeled data as a support set $\mathcal{S}$ from the category set $\mathcal{C}_{novel}$, as well as a certain number of unlabeled data $\mathcal{U}$ acquired also from $\mathcal{C}_{novel}$. Note that, $\mathcal{D}_{novel}$ is disjoint from $\mathcal{D}_{base}$ for generalization test. The task of SSFSL is to learn a robust classifier $f(\cdot; \theta)$ based on both $\mathcal{S}$ and $\mathcal{U}$ for making predictions on new queries $\mathcal{Q}$ from $\mathcal{D}_{novel}$, where $\mathcal{D}_{base}$ is utilized as auxiliary data.

**Setting**    Regarding the *basic semi-supervised few-shot classification* setting, it generally faces the $N$-way-$K$-shot problem, where only $K$ labeled data from $\mathcal{S}$ and $U$ unlabeled data from $\mathcal{U}$ per class are available to learn an $N$-way classifier. In this setting, queries in $\mathcal{Q}$ are treated independently of each other, and are not observed in $\mathcal{U}$. It is referred to as *inductive inference*.

For another important setting in SSFSL, *i.e.*, *transductive inference*, the query set $\mathcal{Q}$ is observed also during training and joint with $\mathcal{U}$.

## 3.2 MUSIC: A Simple Method of sUcceSIve exClusions for SSFSL

The basic idea of our MUSIC is to augment the few-shot labeled set (the support set) $\mathcal{S}$ by predicting "negative" (*i.e.*, "saying *not* belonging to") pseudo-labels to unlabeled data $\mathcal{U}$, particularly for such label-constrained scenarios.

Given an image $I$, we can obtain its representation by training a deep network $F(\cdot; \Theta)$ based on auxiliary data $\mathcal{D}_{base}$:

$$\mathbf{x} = F(I; \Theta) \in \mathbb{R}^d , \tag{1}$$

where $\Theta$ is the parameter of the network. After that, $F(\cdot; \Theta)$ is treated as a general feature embedding function for other images and $\Theta$ is also fixed [31]. Then, considering the task of $c$-class classification, the aforementioned classifier $f(\cdot; \theta)$ maps the input space to a $c$-dimensional score space as

$$\mathbf{p} = \texttt{softmax}(f(\mathbf{x}; \theta)) \in \mathbb{R}^c , \tag{2}$$

where $\mathbf{p}$ is indeed the predicted probability score belonging to the $c$-dimensional simplex $\Delta^{c-1}$, $\texttt{softmax}(\cdot)$ is the softmax normalization, and $\theta$ is the parameter. In SSFSL, $\theta$ is randomly initialized and fine-tuned only by $NK$ labeled data in $\mathcal{S}$ by the cross-entropy loss:

$$\mathcal{L}(f, \mathbf{y}) = -\sum_k y_k \log p_k , \tag{3}$$

where $\mathbf{y} \in \mathbb{R}^c$ is a one-hot vector denoted as the ground-truth label w.r.t. $\mathbf{x}$, and $y_k$ and $p_k$ is the $k$-th element in $\mathbf{y}$ and $\mathbf{p}$, respectively.

To augment the limited labeled data in $\mathcal{S}$, we then propose to predict unlabeled images (*e.g.*, $I^u$) in $\mathcal{U}$ with pseudo-labels from an indirect learning perspective, *i.e.*, excluding negative labels. In concretely, regarding a conventional classification task, the ground-truth $y_k = 1$ represents that its data $\mathbf{x}$ belongs to class $k$, which can be also termed as *positive learning*. In contrast, we hereby denote another one-hot vector $\overline{\mathbf{y}} \in \mathbb{R}^c$ as the counterpart to be the complementary label [10, 12], where $\overline{y}_k = 1$ means that $\mathbf{x}$ does not belong to class $k$, aka *negative learning*. Due to the quite limited labeled data in few-shot learning scenarios, the classifier $f(\cdot; \theta)$ is inaccurate to assign correct positive labels to $I^u$. On the contrary, however, it could be relatively easy and accurate to give such a *negative* pseudo-label to describe that $I^u$ is not from class $k$ by assigning $\overline{y}_k^u = 1$. Therefore, we realize such an idea of "exclusion" by obtaining the most confident negative pseudo-label based on the class having the lowest probability score. The process is formulated as:

$$\overline{y}_k^u = \begin{cases} 1 & \text{if} \quad k = \arg\min(\mathbf{p}^u) \quad \text{and} \quad p_k^u \leq \delta \\ \text{rejection} & \text{otherwise} \end{cases} , \tag{4}$$

where $\mathbf{p}^u$ represents the prediction probability w.r.t. $I^u$, and $\delta$ is a reject option to ensure that there is sufficiently strong confidence to assign pseudo-labels. While if all $p_k^u$ are larger than $\delta$, no negative pseudo-labels are returned for $I^u$ in this iteration.

Thus, after obtaining samples and negative pseudo-label pairs $(I^u, \overline{\mathbf{y}}^u)$, $f(\cdot; \theta)$ can be updated by

$$\mathcal{L}(f, \overline{\mathbf{y}}^u) = -\sum_k \overline{y}_k^u \log(1 - p_k^u) . \tag{5}$$

In the next iteration, we exclude the $k$-th class, *i.e.*, the negative pseudo-label in the previous iteration, from the remaining candidate classes. After that, the updated classifier is employed to give the probability score $\mathbf{p}_{\backslash k}^u \in \mathbb{R}^{c-1}$ of $I^u$, without considering class $k$. The similar pseudo-labeling process is conducted in a successive exclusion manner until all negative pseudo-labels are predicted according to Eqn. (4), or no negative pseudo-labels are able to be predicted with a strong confidence.

Finally, in the last iteration, for those samples in $\mathcal{U}$ whose negative labels are all labeled, their positive pseudo-labels are naturally available. We can further update the classifier by following Eqn. (3) based on the final positive labels. Then, the updated classifier $f(\cdot; \theta)$ is ready for predicting $\mathcal{Q}$ as evaluation.

Moreover, to further improve the probability confidence and then promote pseudo-labeling, we propose to equip a minimum-entropy loss (MinEnt) upon $\mathbf{p}^u$ by optimizing the following objective:

$$\mathcal{L}(f, \mathbf{p}^u) = -\sum_k p_k^u \log p_k^u . \tag{6}$$

**Algorithm 1** Pseudo-code of the proposed MUSIC

```
# f: a classifier, cf. Eqn. (2) of the paper
# δ: a reject option to select the negative label, cf. Eqn. (4) of the paper
# c: the number of classes
# Position: a list to record the label which has been selected as the negative label in each iteration
# S, U: embeddings of the support and unlabeled set which have been extracted by the pre-trained CNN
model (|S|=L, |U|=M)

begin:
    logits ← f(S) # support logits (L, c)
    loss ← CELoss(logits, targets) # CrossEntropy

    while True:
        # negative logits and negative label (M)
        neg_logits, neg_label ← get_neg_samples(Position, f, U, δ)
        if len(neg_label)==0:break # the condition to stop the iterations
        # NegCrossEntropy loss, cf. Eqn. (5); Minimum-Entropy loss, cf. Eqn. (6) of the paper
        loss ← NegCELoss(neg_logits, neg_label) + MiniEntropy(neg_logits)
    end

    pos_logits, pos_label ← get_pos_samples(Position)
    loss ← CELoss(pos_logits, pos_label) + MiniEntropy(pos_logits)
end
```

It could sharp the distribution of $\mathbf{p}^u$ and discriminate the confidence of both positive and negative labels. Algorithm 3.2 provides the pseudo-code of our MUSIC.

## 4 Experiments

### 4.1 Datasets and Empirical Settings

We conduct experiments on four widely-used few-shot learning benchmark datasets for general object recognition and fine-grained classification, including *miniImageNet* [25], *tieredImageNet* [26], *CIFAR-FS* [2] and *CUB* [34]. Specifically, *miniImageNet* consists of 100 classes with 600 samples of $84 \times 84$ resolution per class, which are selected from ILSVRC-2012 [27]. *tieredImageNet* is a larger subset from ILSVRC-2012 with 608 classes in a man-made hierarchical structure, where its samples are also of $84 \times 84$ image resolution. *CIFAR-FS* is a variant of CIFAR-100 [13] with low resolution, which has 100 classes and each of them has 600 samples of $32 \times 32$ size. Regarding *CUB*, it is a fine-grained classification dataset of 200 different bird species with 11,788 images in total.

For fair comparisons, we obey the protocol of data splits in [9, 15, 36] to train the feature embedding function and conduct experiments for evaluations in SSFSL. We choose the commonly used ResNet-12 [7] as the backbone network, and the network configurations are followed [9, 15, 36]. For pre-training, we just follow the same way of [38] to pre-train the network, but do not use any pseudo labels during pre-training. For optimization, Stochastic Gradient Descent (SGD) with momentum of 0.9 and weight decay of $5 \times 10^{-4}$ is adopted as the optimizer to train the feature extractor from scratch. The initial learning rate is 0.1, and decayed as $6 \times 10^{-3}$, $1.2 \times 10^{-3}$ and $2.4 \times 10^{-4}$ after 60, 70 and 80 epochs, by following [38]. Regarding the hyper-parameters in MUSIC, the reject option $\delta$ in Eqn. (4) is set to $\frac{1}{c}$ and the trade-off parameter over Eqn. (6) is set to 1 as default for all experiments and iterations, which shows its practicality and non-tricky. During evaluation, the last layer of pre-trained model is replaced by an $\ell_2$-normalization layer and a $c$-dimensional fully connected layer as the classifier. We also use SGD for optimization. Our MUSIC and all baselines are evaluated over 600 episodes with 15 test samples in each class. All experiments are conducted by MindSpore with a GeForce RTX 3060 GPU.

### 4.2 Main Results

We report the empirical results in the following four setups. All results are the average accuracy and the corresponding 95% confidence interval over the 600 episodes are also conducted.

**Basic semi-supervised few-shot setup**  We compare our MUSIC with state-of-the-art methods in the literature in Table 1. As shown, our simple approach outperforms the competing methods of both generic few-shot learning and semi-supervised few-shot learning by a large margin across different few-shot tasks over all the datasets. Beyond that, we also report the results of solely using

Table 1: Comparisons of 5-way few-shot classification *with the basic semi-supervised few-shot setup*. These results are performed with 30/50 unlabeled samples for 5-way-1-shot/5-way-5-shot, respectively. The light blue blocks represent that these methods are tested in the inductive setup, and the light yellow blocks are tested in the basic semi-supervised setup. The highest accuracy is marked in **red**, and the second highest accuracy is in **blue**.

| METHOD | BACKBONE | miniImageNet | | tieredImageNet | | CIFAR-FS | | CUB | |
|---|---|---|---|---|---|---|---|---|---|
| | | 1-shot | 5-shot | 1-shot | 5-shot | 1-shot | 5-shot | 1-shot | 5-shot |
| MatchingNet [33] | 4 CONV | 43.56 | 55.31 | – | – | – | – | – | – |
| MAML [4] | 4 CONV | 48.70 | 63.11 | 51.67 | 70.30 | 58.90 | 71.50 | 54.73 | 75.75 |
| ProtoNet [29] | 4 CONV | 49.42 | 68.20 | 53.31 | 72.69 | 55.50 | 72.00 | 50.46 | 76.39 |
| LEO [28] | WRN-28-10 | 61.76 | 77.59 | 66.33 | 81.44 | – | – | – | – |
| CAN [8] | ResNet-12 | 63.85 | 79.44 | 69.89 | 84.23 | – | – | – | – |
| DeepEMD [45] | ResNet-12 | 65.91 | 82.41 | 71.16 | 86.03 | 74.58 | 86.92 | 75.65 | 88.69 |
| FEAT [43] | ResNet-12 | 66.78 | 82.05 | 70.80 | 84.79 | – | – | 73.27 | 85.77 |
| RENet [11] | ResNet-12 | 67.60 | 82.58 | 71.61 | 85.28 | 74.51 | 86.60 | 82.85 | 91.32 |
| FRN [40] | ResNet-12 | 66.45 | 82.83 | 72.06 | 86.89 | – | – | 83.55 | 92.92 |
| COSOC [21] | ResNet-12 | 69.28 | 85.16 | 73.57 | 87.57 | – | – | – | – |
| SetFeat [1] | ResNet-12 | 68.32 | 82.71 | 73.63 | 87.59 | – | – | 79.60 | 90.48 |
| MCL [20] | ResNet-12 | 69.31 | 85.11 | 73.62 | 86.29 | – | – | 85.63 | 93.18 |
| STL DeepBDC [41] | ResNet-12 | 67.83 | 85.45 | 73.82 | 89.00 | – | – | 84.01 | **94.02** |
| TPN [19] | 4 CONV | 52.78 | 66.42 | 55.74 | 71.01 | – | – | – | – |
| TransMatch [44] | WRN-28-10 | 60.02 | 79.30 | 72.19 | 82.12 | – | – | – | – |
| LST [17] | ResNet-12 | 70.01 | 78.70 | 77.70 | 85.20 | – | – | – | – |
| EPNet [23] | ResNet-12 | 70.50 | 80.20 | 75.90 | 82.11 | – | – | – | – |
| ICI [36] | ResNet-12 | 69.66 | 80.11 | 84.01 | 89.00 | 76.51 | 84.32 | 89.58 | 92.48 |
| iLPC [15] | ResNet-12 | 70.99 | 81.06 | 85.04 | 89.63 | 78.57 | 85.84 | 90.11 | – |
| PLCM [9] | ResNet-12 | 72.06 | 83.71 | 84.78 | 90.11 | 77.62 | 86.13 | – | – |
| **Ours** | ResNet-12 | **74.96** | **85.99** | **85.40** | **90.79** | **78.96** | **87.25** | **90.76** | **93.27** |
| **Ours (only neg)** | ResNet-12 | 73.86 | 85.11 | 84.91 | 90.29 | 78.26 | 86.53 | 89.91 | 92.46 |
| **Ours (only pos)** | ResNet-12 | **74.44** | **85.86** | **85.33** | **90.62** | **78.81** | **87.11** | **90.27** | 93.11 |

Table 2: Comparisons of 5-way few-shot classification *with the transductive setup*. The highest accuracy is marked in **red**, and the second highest accuracy is in **blue**.

| METHOD | BACKBONE | miniImageNet | | tieredImageNet | | CIFAR-FS | | CUB | |
|---|---|---|---|---|---|---|---|---|---|
| | | 1-shot | 5-shot | 1-shot | 5-shot | 1-shot | 5-shot | 1-shot | 5-shot |
| TPN [19] | 4 CONV | 55.51 | 69.86 | 59.91 | 73.30 | – | – | – | – |
| EPNet [23] | ResNet-12 | 66.50 | 81.06 | 76.53 | 87.32 | – | – | – | – |
| ICI [36] | ResNet-12 | 66.80 | 79.26 | 80.79 | 87.92 | 73.97 | 84.13 | 88.06 | 92.53 |
| iLPC [15] | ResNet-12 | 69.79 | 79.82 | **83.49** | 89.48 | 77.14 | 85.23 | 89.00 | 92.74 |
| PLCM [9] | ResNet-12 | 70.92 | 82.74 | 82.61 | 89.47 | – | – | – | – |
| **Ours** | ResNet-12 | **72.01** | **83.49** | **83.57** | **89.81** | **77.56** | **85.49** | **89.40** | **92.91** |
| **Ours (only neg)** | ResNet-12 | 71.46 | 83.04 | 83.20 | 89.33 | 77.26 | 85.10 | 88.73 | 92.51 |
| **Ours (only pos)** | ResNet-12 | **71.83** | **83.31** | 83.44 | **89.57** | **77.42** | **85.33** | **89.31** | **92.78** |

the pseudo-labeled negative or positive samples generated by our MUSIC, which is denoted by "Ours (only neg)" or "Ours (only pos)" in that table. It is apparent to observe that even only using negative pseudo-labeling, MUSIC can still be superior to other existing FSL methods. Moreover, compared with the results of only using positive pseudo-labeling, the results of only using negative are worse. It reveals that accurate positive labels still provide more information than negative labels [10].

**Transductive semi-supervised few-shot setup**   In the transductive setup, it is available to access the query data in the inference stage. We also perform experiments in such a setup and report the results in Table 2. As seen, our approach can still achieve the optimal accuracy on all the four datasets, which justifies the effectiveness of our MUSIC. Regarding the comparisons between (only) using negative and positive pseudo-labels, it has similar observations as those in Table 1.

Table 3: Comparisons of 5-way few-shot classification *with the distractive semi-supervised setup.*

| METHOD | miniImageNet | | tieredImageNet | |
|---|---|---|---|---|
| | 1-shot | 5-shot | 1-shot | 5-shot |
| MS $k$-Means [26] | 49.00 | 63.00 | 51.40 | 69.10 |
| TPN [19] | 50.40 | 64.90 | 53.50 | 69.90 |
| TPN with MTL [19] | 61.30 | 72.40 | 71.50 | 82.70 |
| LST [17] | 64.10 | 77.40 | 73.50 | 83.40 |
| EPNet [23] | 64.70 | 76.80 | 72.20 | 82.10 |
| PLCM [9] | 68.50 | 80.20 | 79.10 | 87.80 |
| **Ours** | **68.62** | **80.67** | **79.69** | **88.50** |

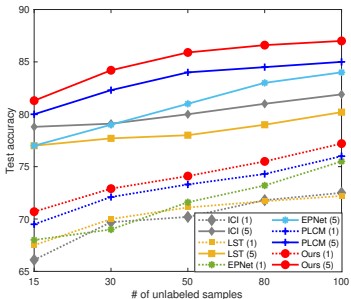

Figure 2: Comparison results of *varied unlabeled samples* on *miniImageNet*. The number in the brackets of the legend represents $K$-shot in SSFSL.

Table 4: Comparisons of 5-way few-shot classification *with different orders of negative and positive pseudo-labeling w.r.t. unlabeled data.*

| SETTINGS | miniImageNet | | CUB | |
|---|---|---|---|---|
| | 1-shot | 5-shot | 1-shot | 5-shot |
| neg $\to$ pos $\to \cdots$ | **74.77** | **85.43** | **90.47** | **92.59** |
| pos $\to$ neg $\to \cdots$ | 74.54 | 85.09 | 90.29 | 92.24 |

Table 5: Comparisons of 5-way few-shot classification *without / with the minimum-entropy loss (MinEnt)*, cf. Eqn. (6).

| SETTINGS | miniImageNet | | CUB | |
|---|---|---|---|---|
| | 1-shot | 5-shot | 1-shot | 5-shot |
| Ours (w/o MinEnt) | 74.73 | 85.74 | 90.45 | 92.98 |
| Ours (w/ MinEnt) | **74.96** | **85.99** | **90.76** | **93.27** |

**Distractive semi-supervised few-shot setup**    In real applications, it might not be realistic to collect a clean unlabeled set without mixing any data of other classes. To further validate the robustness of MUSIC, we conduct experiments with the distractive setup, *i.e.*, the unlabeled set contains distractive classes which are excluded in the support set. In that case, positive pseudo-labels are more prone to error, while negative pseudo-labels have a much lower risk of error. Table 3 presents the comparison results and shows that our approach can perform as the best solution in all distractive semi-supervised few-shot classification tasks.

**Variety-unlabeled semi-supervised few-shot setup**    In order to analyze the performance in the case of different unlabeled samples, we perform our MUSIC under the variety-unlabeled semi-supervised setup and compare with state-of-the-arts, *e.g.*, ICI [36], LST [17] and PLCM [9]. As shown in Figure 2, our approach significantly outperforms over these methods in different $K$-shot tasks of SSFSL. It further validates the effectiveness and generalization ability of our MUSIC.

### 4.3   Ablation Studies and Discussions

We hereby analyze and discuss our MUSIC approach by answering the following questions based on ablation studies on two datasets, *i.e.*, *miniImageNet* and *CUB*.

**Will negative pseudo-labels be easier to predict under SSFSL than positive ones?**    As assumed previously, in such an extremely label-constrained scenario, *e.g.*, 1-shot learning, it might be hard to learn an accurate classifier for correctly predicting positive pseudo-labels. In this sub-section, we conduct ablation studies by alternatively performing negative and positive pseudo-labeling to verify this assumption. In Table 4, different settings denote different orders of negative and positive pseudo-labeling in SSFSL. For example, "neg $\to$ pos $\to \cdots$" represents that we firstly obtain negative pseudo-labels by our MUSIC (without using the final positive labels) and update the model, and then we obtain positive pseudo-labels[3] and update model, and so on. Regarding the iteration time, it is relevant to the number of $K$ in the $K$-way classification. In concretely, for 5-way classification, our MUSIC returns the most confident negative pseudo-label in the current iteration and excludes it for the next iteration. Thus, after four times of "neg $\to$ pos", all negative pseudo-labelings are finished and the results can be reported. Similarly, "pos $\to$ neg $\to \cdots$" means that we get the positive pseudo-labels first, followed by the negative ones. As the results shown in Table 4, we can see that obtaining negative pseudo-labels first obviously achieves better results than positive first, which

---

[3]The method of positive pseudo-labeling here is a baseline solution, which trains a classifier with cross-entropy and obtains the positive pseudo-label by the highest logits above a certain threshold (*e.g.*, 0.7).

shows that labeling negative pseudo-labels first can lay a better foundation for model training, and further answers the question in this sub-section as "*YES*".

**Is the minimum-entropy loss effective?** In our MUSIC, to further improve the probability confidence and then promote pseudo-labeling, we equip the minimum-entropy loss (MinEnt). We here test its effectiveness and report the results in Table 5. It can be found that training with MinEnt (*i.e.*, the proposed MUSIC) brings 0.2∼0.3% improvements over training without MinEnt in SSFSL.

**Is the reject option $\delta$ effective?** We hereby verify the effectiveness and necessity of the reject option $\delta$ in MUSIC. The $\delta$ in our approach acts as a safeguard to ensure that the obtained negative pseudo-labels are as confident as possible. We present the results in Table 6, and can observe that MUSIC with $\delta$ achieves significantly better few-shot classifi-

Table 6: Comparisons of 5-way few-shot classification *without / with the reject option $\delta$*, cf. Eqn. (4).

| SETTINGS | miniImageNet | | CUB | |
|---|---|---|---|---|
| | 1-shot | 5-shot | 1-shot | 5-shot |
| Ours (w/o $\delta$) | 74.04 | 85.31 | 90.44 | 92.87 |
| Ours (w/ $\delta$) | **74.96** | **85.99** | **90.76** | **93.27** |

cation accuracy than MUSIC without $\delta$. Additionally, even without $\delta$, our approach can still perform well, *i.e.*, the results being comparable or even superior to the results of state-of-the-arts.

**What is the effect of iteration manner in our MUSIC?** As aforementioned, our approach works as a successive exclusion manner until all negative pseudo-labels are predicted, and eventually obtaining positive pseudo-labels. As pseudo-labeling conducting, it is interesting to investigate how the performance changes as the iteration progresses. We report the corresponding results in Figure 3. As shown, on each task of these two datasets, our approach all shows a relatively stable growth trend, *i.e.*, 0.5∼2% improvements over the previous iteration.

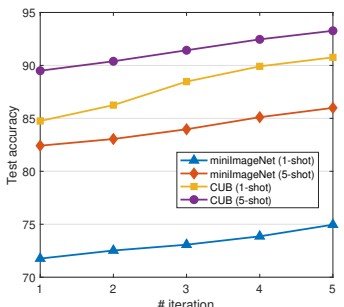

Figure 3: Our results of 5-way few-shot classification with iteration increasing.

**What is the performance of pseudo-labeling in MUSIC?** In this sub-section, we explicitly investigate the error rates of both negative and positive pseudo-labels predicted by our approach. We take 5-way-5-shot classification on *miniImageNet* and *CUB* as examples, and first present the pseudo-labeling error rates of negative labels in Table 7. Since the task is 5-way prediction, there are totally four iterations of negative pseudo-labeling in MUSIC reported in that table. Except for error rates, we also detailedly report the number of wrong labeled samples in each iteration, as well as the total number of labeled samples. Note that, in the third and forth iterations of negative pseudo-labeling, the total number of labeled samples are less than the number of unlabeled data (*i.e.*, 250), which is due to the reject option in MUSIC. That is to say, those samples cannot be pseudo-labeled with a strongly high confidence. Meanwhile, we also see that, as the pseudo-labeling progresses, the error rates slowly increase, but the final error rate of negative labeling is still no higher than 6.7%. It demonstrates the effectiveness of our approach from a straightforward view.

On the other side, Table 8 compares the positive pseudo-labeling error rates, and also reports the proportion of labeled samples in the total number of unlabeled samples. Regarding ICI [36] and iLPC [15], although they designed tailored strategies to ensure the correctness of pseudo-labels, *e.g.*, instance credibility inference [36] and label cleaning [15], these methods still have high pseudo-labeling error rates (over 25%). Compared with them, our approach has significantly low error rates, *i.e.*, about 10%. Meanwhile, we also note that our MUSIC only predicts about 80% of the unlabeled data, which can be regarded to be relatively conservative. However, it reveals that our

Table 7: Pseudo-labeling error rates of *negative* labels of each iteration in 5-way-5-shot classification.

| ITERATION | 1 | 2 | 3 | 4 |
|---|---|---|---|---|
| *miniImageNet* | 0.43% (1.1/250) | 1.17% (2.9/250) | 2.93% (7.3/249) | 4.15% (8.2/198) |
| *CUB* | 0.69% (1.3/195) | 1.83% (3.6/195) | 4.47% (8.5/190) | 6.63% (10/152) |

Table 8: Pseudo-labeling error rates and proportion of *positive* labels in 5-way-5-shot classification.

| METRIC | DATASET | ICI [36] | iLPC [15] | Ours (w/o $\delta$) | **Ours** |
|---|---|---|---|---|---|
| ERROR RATE | *miniImageNet* | 24.72% (61.8/250) | 23.92% (59.8/250) | 14.92% (37.3/250) | **9.09% (18.0/198)** |
| | *CUB* | 29.03% (56.6/195) | 26.10% (50.9/195) | 20.00% (39.0/195) | **11.91% (18.1/152)** |
| PROPORTION | *miniImageNet* | 100% | 100% | 100% | 79.20% |
| | *CUB* | 100% | 100% | 100% | 77.95% |

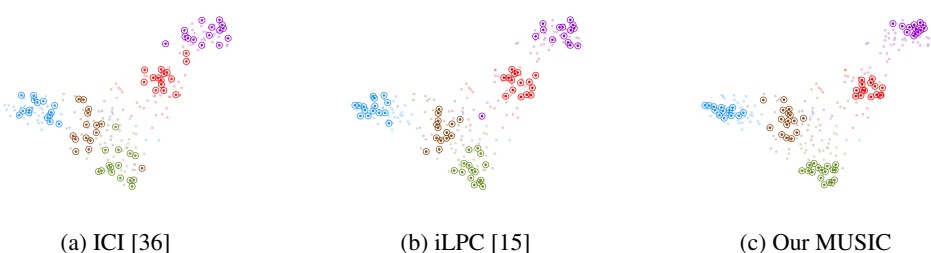

(a) ICI [36]  (b) iLPC [15]  (c) Our MUSIC

Figure 4: $t$-SNE [32] visualization of samples for the 5-way 5-shot 50-unlabeled task. Different colors denote different classes. For ICI and iLPC, the circled data points represent the selected samples. For our MUSIC, the circled are the samples associated with high confidence of positive pseudo-labels in the final iteration.

approach still has a large room for performance improvement. Moreover, Table 8 also shows that, even if our approach removes the reject option strategy, its error rates are still lower than those of state-of-the-arts.

Additionally, we visualize the positive pseudo-labels with high confidence by $t$-SNE [32] in Figure 4. Compared with these methods, we can obviously find that the positive samples with high confidence predicted by our MUSIC are both more centralized and distinct. This also explains the satisfactory performance of our approach when using the positive pseudo-labels and using the positive alone (cf. Table 1 and Table 2) from the qualitative perspective.

**Are the pseudo-labels of MU-SIC a balanced distribution?** In this sub-section, we are still interested in investigating what kind of data distribution the pseudo-labeled samples are to further analyze how our ap-

Table 9: The averaged number of negative (positive) pseudo-labeled samples not-belonging (belonging) to different classes in our MUSIC.

| CLASS INDEX | 1 | 2 | 3 | 4 | 5 |
|---|---|---|---|---|---|
| # of neg. pseudo-labeling | 49.85 | 49.74 | 49.93 | 50.19 | 50.30 |
| # of pos. pseudo-labeling | 40.13 | 41.13 | 40.21 | 39.89 | 39.99 |

proach works well. As shown in Table 9, we present the averaged number of both negative and positive pseudo-labeled samples in all 600 episodes of 5-way-5-shot classification tasks on *mini-ImageNet*. It is apparent to see that the pseudo-labeled samples present a very clearly balanced distribution, which aids in the modeling of classifiers across different classes in SSFSL.

## 5 Conclusion

In this paper, we dealt with semi-supervised few-shot classification by proposing a simple but effective approach, termed as MUSIC. Our MUSIC worked in a successive exclusion manner to predict negative pseudo-labels with much confidence as possible in the extremely label-constrained tasks. After that, models can be updated by leveraging negative learning based on the obtained negative pseudo-labels, and continued negative pseudo-labeling until all negative labels were returned. Finally, combined with the incidental positive pseudo-labels, we augmented the small support set of labeled data for evaluation in SSFSL. In experiments, comprehensive empirical studies validated the effectiveness of MUSIC and revealed its working mechanism. In the future, we would like to investigate the theoretical analyses about our MUSIC in terms of its convergence and estimation error bound, as well as how it performing on traditional semi-supervised learning tasks.

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
