# OpenReview forum: "An Embarrassingly Simple Approach to Semi-Supervised Few-Shot Learning"
_NeurIPS.cc/2022/Conference — NeurIPS 2022 Accept_

### Official Review · Reviewer_nug7 · 2022-07-09

**Rating:** 6
**Confidence:** 3
**Soundness:** 3 good
**Presentation:** 2 fair
**Contribution:** 3 good

**Summary:**

This work applies negative learning to the problem of semi-supervised few-shot learning. It uses negative pseudo-label to gradually get rid of unlikely predictions. It also minimizes the entropy of the predicted probability of unlabeled target domain data to promote pseudo-labeling.

**Questions:**

1. The entropy-loss in (6) seems to share a similar spirit with pseudo-labeling (also known as soft pseudo-labeling in other places). What if you apply only (6) and do not use (5) at all?

2. The author of the baseline work [38] also uses pseudo labels, and they rank pseudo labels in a sophisticated way. When you say pretraining the network as in [38], do you mean you use their method to pre-train the model, or do you mean you pre-train it in the same way but do not use pseudo labels at all during pre-training?

**Limitations:**

The author needs to compare the baseline work [38] clearly to show a more convincing result.

**Strengths And Weaknesses:**

Strength:
In general, the paper is easy to follow and highlights the key point clearly.

Weakness:
1. The major problem of this work lies in the experiment section. The author mentioned that this work pretraining the network as in previous work [38], so it should be the baseline of this work. However, the table does not include that work [38] at all. Furthermore, comparing Table 2 in the baseline work [38] and Table 1&2 in this work, we can find that the baseline work [38] actually performs better in some settings.

2. While negative learning is the key point of this work, as shown in Table 1&2, the improvement from using only positive pseudo-label to using both negative and positive pseudo-labels seems trivial. Since pseudo-label itself is a well-known idea, the novelty will be very limited if the negative pseudo-labels do not really make a difference.

---

> ### Author Response · Authors · 2022-07-30
> **Thank you for the comments. Below please find our responses to some specific comments.**
>
> *Comment_1: When say pretraining the network as in [38], do you mean you use their method to pre-train the model, or do you mean you pre-train it in the same way but do not use pseudo labels at all during pre-training?*
>
> Response_1: When pre-training the network as in [38], we only pre-trained it in the same way but do NOT use any pseudo labels at all during pre-training. We will clarify this statement in the final version.
> ***
> *Comment_2: The author needs to compare the baseline work [38] clearly to show a more convincing result.*
>
> Response_2: We would like to point out that [38] employs logistic regression (LR) as the classifier, while our MUSIC uses a linear fully connected (FC) layer as the classifier. Many state-of-the-art SSFSL methods have chosen LR as the classifier for better accuracy, e.g., [38, 15, 31, 36]. Thus, for fair and comprehensive comparisons with [38] by following your comment, we change our original FC classifier to LR and further conduct experiments on the distractive semi-supervised setup (cf. Table 3 of the paper). For [38] with the distractive setup, we use the source codes released by its authors for experiments. The results are shown in the following tables.
>
> Response-Table-1: _Comparisons of the basic semi-supervised few-shot setup._
> ||_miniImageNet_|_miniImageNet_|_tieredImageNet_|_tieredImageNet_|_CIFAR-FS_|_CIFAR-FS_|_CUB_|_CUB_|
> |:----------------------:|:--------------:|:--------------:|:----------------:|:----------------:|:----------:|:----------:|:---------:|:---------:|
> ||1-shot|5-shot|1-shot|5-shot|1-shot|5-shot|1-shot|5-shot|
> |[38]|73.12|83.28|78.99|86.76|**80.74**|87.16|**92.12**|94.52|
> |**Ours (Original FC)**|74.96|85.99|85.40|90.79|78.96|87.25|90.76|93.27|
> |**Ours (LR)**|**75.60**|**86.61**|**86.03**|**91.23**|79.67|**87.97**|91.69|**94.60**|
>
> Response-Table-2: _Comparisons of the transductive setup._
> ||_miniImageNet_|_miniImageNet_|_tieredImageNet_|_tieredImageNet_|_CIFAR-FS_|_CIFAR-FS_|_CUB_|_CUB_|
> |:----------------------:|:--------------:|:--------------:|:----------------:|:----------------:|:----------:|:----------:|:---------:|:---------:|
> ||1-shot|5-shot|1-shot|5-shot|1-shot|5-shot|1-shot|5-shot|
> |[38]|72.39|83.27|77.48|86.84|**79.19**|**86.66**|**90.89**|94.36|
> |**Ours (Original FC)**|72.01|83.49|83.57|89.81|77.56|85.49|89.40|92.91|
> |**Ours (LR)**|**72.69**|**84.14**|**84.18**|**90.45**|78.24|86.18|90.23|**94.38**|
>
> Response-Table-3: _Comparisons of the distractive semi-supervised setup._
> ||_miniImageNet_|_miniImageNet_|_tieredImageNet_|_tieredImageNet_|
> |:----------------------:|:--------------:|:--------------:|:----------------:|:----------------:|
> ||1-shot|5-shot|1-shot|5-shot|
> |[38]|68.09|77.71|73.47|82.09|
> |**Ours (Original FC)**|68.62|80.67|79.69|88.50|
> |**Ours (LR)**|**69.14**|**81.20**|**80.23**|**89.11**|
>
> As shown, in most cases (15 wins and 5 losses), our MUSIC with LR outperforms [38] by a large margin, especially for the two most popular used benchmark datasets, i.e., _miniImageNet_ and _tieredImageNet_. In addition, even if in several cases our method is worse than [38], the accuracy drop is basically no more than 1%. More importantly, for the realistic setting, i.e., distractive SSFSL, our methods have great improvements over [38], which reveals its good robustness.
> ***
> *Comment_3: Improvement from using only positive pseudo-label to using both negative and positive pseudo-labels seems trivial.*
>
> Response_3: We would like to emphasize that what our work proposes is a novel way of annotating pseudo-labels based on negative learning, rather than simply emphasizing the effect of negative pseudo labels. In our method, both positive and negative pseudo-labels are obtained by our negative learning-based strategy. Compared with previous works, our MUSIC with only positive pseudo-labels (i.e., "Ours (only pos)" in Table 1 & 2 of the paper) can still outperform them, which can validate the effectiveness of the obtained positive pseudo-labels by our negative learning-based pseudo-labeling strategy. Moreover, as noticed by the other three reviewers (cf. the paper strengths of their reviews), they also praised our novelty and contributions of our negative pseudo-labeling strategy.
> ***
> *Comment_4: What if you apply only (6) and do not use (5)?*
>
> Response_4: We conduct experiments by only applying (6) and do not use (5), and report the results of the basic semi-supervised few-shot setup as follows.
> ||_miniImageNet_|_miniImageNet_|_tieredImageNet_|_tieredImageNet_|_CIFAR-FS_|_CIFAR-FS_|_CUB_|_CUB_|
> |:--------:|:--------------:|:--------------:|:----------------:|:----------------:|:----------:|:----------:|:---------:|:---------:|
> ||1-shot|5-shot|1-shot|5-shot|1-shot|5-shot|1-shot|5-shot|
> |Only (6)|72.84|83.21|83.54|88.36|76.83|85.47|89.50|92.15|
> |**Ours**|**74.96**|**85.99**|**85.40**|**90.79**|**78.96**|**87.25**|**90.76**|**93.27**|
>
> As shown, although applying only (6) can still work, its results are significantly worse than the results of our MUSIC.

---

> > ### Comment · Reviewer_nug7 · 2022-08-05
> > **Review to comments**
> >
> > Comment_1: Thanks for clarification
> >
> > Comment_2: Thanks for the detailed result. However, it is unclear to me why you want to emphasize that you use FC layers rather than LR. Is there any drawback using LR?
> >
> > Comment_3: For your "only positive pseudo-labels", do you mean you use loss (3) and ignore (5),  but would still exclude the k-th class, i.e., the negative pseudo-label in the previous iteration?
> >
> > Comment_4: Thanks for clarification

---

> > > ### Author Response · Authors · 2022-08-05
> > > **Responses to Reviewer nug7**
> > >
> > > Thank you so much for the feedback. Below please find our responses to the additional comments.
> > > ***
> > > *Comment_2: It is unclear to me why you want to emphasize that you use FC layers rather than LR. Is there any drawback using LR?*
> > >
> > > Response_2: There is no drawback using LR, on the contrary, LR will bring higher accuracy. Our previous response just wanted you to notice that the final classifier used by [38] and ours is different (or even unfair), and this classifier has an impact on the classification accuracy of SSFSL tasks. In fact, in our original experiments, we chose to use FC only because it can be more easily coupled with the overall network training. In the rebuttal stage, thanks to your suggestion that we "need to compare the baseline work [38] clearly to show a more convincing result", we noticed that [38] was using the off-the-shelf classifier LR for the final classification. Moreover, we also find that LR leads to better classification accuracy as shown in these works, e.g., [38, 15, 31, 36]. Therefore, we replace the original FC classifier with LR for fair and sufficient experimental comparisons with [38], whose results are as presented in these tables above. As shown, our MUSIC with LR does further improve the SSFSL accuracy over our MUSIC with FC, which also significantly outperforms [38] in most cases (15 wins and 5 losses), especially for the realistic distractive SSFSL setting.
> > >
> > > ***
> > > *Comment_3: For your "only positive pseudo-labels", do you mean you use loss (3) and ignore (5), but would still exclude the $k$-th class, i.e., the negative pseudo-label in the previous iteration?*
> > >
> > > Response_3: Yes. We will clarify this statement in the final version. Thank you.

---

> > > > ### Comment · Reviewer_nug7 · 2022-08-07
> > > > **Review to comments**
> > > >
> > > > Thanks to the author for the clarification. My original low rating comes from two aspects:
> > > > 1.  It was unclear about what pre-training the network as in [38] means and made it suspicious as an incremental work to [38]
> > > > 2.  I misinterpret "using only positive pseudo-labels" as "not using negative pseudo-label at all, so not excluding the k-th class", which makes it looks like a basic method with pseudo-label
> > > >
> > > > The detailed responses from the author alleviate these concerns, thus, I will recommend a weak acceptance of the paper instead, but I will suggest the author clarify these points in the main paper.

---

> > > > > ### Author Response · Authors · 2022-08-07
> > > > > **Thank you**
> > > > >
> > > > > Thank you so much for the valuable feedback and your recommendation to accept our work. We will clarify these mentioned points in the final version.

---

### Official Review · Reviewer_46zy · 2022-07-09

**Rating:** 6
**Confidence:** 4
**Soundness:** 4 excellent
**Presentation:** 4 excellent
**Contribution:** 3 good

**Summary:**

This submission proposes a simple yet effective learning method for Semi-Supervised Few-Shot Learning (SSFSL) called MUSIC. Compared to previous methods, the authors propose to learn negative labels first and then focus on positive label learning. The underlying logic is that under few-shot learning scenario, it is easier to exclude negative predicted labels than select positive labels. The experiments show that the simple method achieves state-of-the-art performance on four benchmark datasets.

**Questions:**

Overall I have little confusion for the methodology and technical details. There are two suggestions:

1) In the ablation study, the authors only investigate whether the reject option $\delta$ is effective or not. It would be better to further study what would be the optimal value of $\delta$ and whether this hyper parameter is agnostic to different datasets' distribution.

2) In Table 4, the authors conduct an interesting experiment for the order of negative and positive pseudo labels learning. I am curious what is the optimal number of iterations for the neg -> pos -> neg ... in the MUSIC method, or it is dependent on different datasets?

**Limitations:**

I am basically satisfied with the submission in terms of methodology, motivation and experiments. I provided some suggestions in the above section ("questions") which also contain some constructive suggestions.

**Strengths And Weaknesses:**

+ The authors propose a straightforward learning method for SSFSL. The motivation is clear and technical details are illustrated with sufficient details.

+ The authors conduct experiments on four benchmark datasets and achieves state-of-the-art performance.

+ The authors further dive into different aspects of the MUSIC method and provide ablation study, which is much appreciated.

---

> ### Author Response · Authors · 2022-07-28
> **Thank you for the positive comments. Below please find our responses to some specific comments.**
>
> *Comment_1: In the ablation study, the authors only investigate whether the reject option $\delta$ is effective or not. It would be better to further study what would be the optimal value of $\delta$ and whether this hyper parameter is agnostic to different datasets' distribution.*
>
> Response_1: As stated in the experiments (cf. Ln. 181 of the paper), the reject option $\delta$ is set to $1/c$ (i.e., 0.20) as the default value for all experiments. We had indeed conducted ablation studies of different values of $\delta$ to show its sensitiveness. The results on *miniImageNet* and *CUB* are reported as follows. We can obviously find that different values of $\delta$ are not sensitive and $\delta$ is agnostic to different datasets. We will also add these experiments and discussions in the final version by following your constructive suggestion. Thank you.
> | _Value of $\delta$_     | _miniImageNet_ | _miniImageNet_ | _CUB_    | _CUB_    |
> |:------:|:--------------:|:--------------:|:--------:|:--------:|
> |  $K$-shot    | 1-shot       | 5-shot       | 1-shot | 5-shot |
> | 0.15 | 74.84        | 85.80        | 90.63  | 93.11  |
> | 0.20 | 74.96        | 85.99        | 90.76  | 93.27  |
> | 0.25 | 75.04        | 86.17        | 90.69  | 93.20  |
> | 0.30 | 74.85        | 85.73        | 90.61  | 93.17  |
> ***
> *Comment_2: In Table 4, the authors conduct an interesting experiment for the order of negative and positive pseudo labels learning. I am curious what is the optimal number of iterations for the neg -> pos -> neg ... in the MUSIC method, or it is dependent on different datasets?*
>
> Response_2: It is not dependent on different datasets. In this ablation study, since the task is 5-way few-shot classification, we fix the number of iterations as four times of "neg->pos" (or "pos->neg"). More specifically, at each time/iteration, our MUSIC returns the most confident negative pseudo-label in the current iteration and excludes it for the next iteration. Thus, after four times of negative pseudo-labeling by our MUSIC, all negative labels can be obtained (cf. Ln. 151-152 of the paper). Therefore, the iteration times are relevant to the number of $K$ in the $K$-way classification.

---

### Official Review · Reviewer_QttX · 2022-07-10

**Rating:** 7
**Confidence:** 4
**Soundness:** 4 excellent
**Presentation:** 4 excellent
**Contribution:** 3 good

**Summary:**

The paper proses a simple approach to SSFSL that employs negative label prediction when producing pseudo-labels for unlabelled examples. The model consists of a standard ResNet12 architecture that is pre-trained on the base data following an L2-normalizing single-layer classifier that is fine-tuned based on the support data. The fine-tuning process first updates parameters based on standard cross-entropy loss of the support examples. It then iteratively removes negative pseudo-labels from the unlabelled examples but using a modified cross-entropy negative label predictor loss, and finally, performs updates based on the positive pseudo-labels of the unlabelled set for examples which all negative labels have been identified. Various experiments and ablations studies are reported that demonstrate the efficacy of the approach.

**Questions:**

The choice to decouple the training of the feature extractor from the few-shot learning algorithm itself is an interesting one. It is often seen that end-to-end training of the extractor through episodic procedures where the few-shot updates are also applied results in better performance. More specifically, the updates shown in the PyTorch code block could be applied directly to f and F together where the inputs are just the raw images. Was this something the authors explored? If so, what was the outcome? If not, why not?

**Limitations:**

The authors have adequately addressed technical limitations of the work (although further studies on empirical biases based on data domain would have been interesting to see). However, there is no discussion of potential negative societal impact of the work; in fairness to the authors, this is an algorithmic work and the societal impacts can be speculative at time; but they could benefit from a short discussion of what their method, as an effective SSFSL classifier, can enable in applied industrial settings.

**Strengths And Weaknesses:**

Strengths:
- Paper is overall very well-written.
- Algorithmic choices are well-motivated, backed up by both good intuition and supportive ablation studies.
- Method is simple to understand on the first go but also very empirically powerful as demonstrated by series of experiments.
- Negative labelling is a very interesting insight and can prove consequential specifically in the domain of SSFSL which is very applicable to applied settings where labelling can be expensive but lots of unlabelled data is available.

Weakness:
- There are some typos, such as "detailedly" (261), "can performs" (208), and most important "logits = f.forward(x)" and
"loss = F.nll_loss(F.log_softmax(logits), labels)" in the algorithm blocks where "x" should be "S" and "labels" should be "targets" to my understanding of the procedure
- The algorithm block is in PyTorch which I personally appreciate but can be difficult to navigate if read doesn't have existing PyTorch proficiency; I believe a pseudo-code algorithm block would be more appropriate with the PyTorch code moved to the supplementary material. I recognize that this was done to reinforce the fact that the method can be implemented in a few-lines of code.

Middleground:
- The algorithm is simple and effective; but as a result doesn't contain very significant technical novelty and contribution. That being said, the authors have embraced its simplicity in the language of the paper throughout which addresses this potential problem.

---

> ### Author Response · Authors · 2022-07-28
> **Thank you for the positive comments. Below please find our responses to some specific comments.**
>
> *Comment_1: There are some typos, such as "detailedly" (261), "can performs" (208), and most important "logits = f.forward(x)" and "loss = F.nll_loss(F.log_softmax(logits), labels)" in the algorithm blocks where "x" should be "S" and "labels" should be "targets" to my understanding of the procedure.*
>
> Response_1: Thank you so much for pointing out these issues. We will fix them in the final version.
> ***
> *Comment_2: The algorithm block is in PyTorch which I personally appreciate but can be difficult to navigate if read doesn't have existing PyTorch proficiency; I believe a pseudo-code algorithm block would be more appropriate with the PyTorch code moved to the supplementary material. I recognize that this was done to reinforce the fact that the method can be implemented in a few-lines of code.*
>
> Response_2: Thank you for the constructive suggestion. We will follow the suggestion to polish the algorithm block in the final version.
> ***
> *Comment_3: The choice to decouple the training of the feature extractor from the few-shot learning algorithm itself is an interesting one. It is often seen that end-to-end training of the extractor through episodic procedures where the few-shot updates are also applied results in better performance. More specifically, the updates shown in the PyTorch code block could be applied directly to f and F together where the inputs are just the raw images. Was this something the authors explored? If so, what was the outcome? If not, why not?*
>
> Response_3: In fact, we follow [31] of the paper to decouple the training of the feature extractor from the few-shot learning algorithm itself, and [31] empirically encourages to not fine-tune the feature extractor (aka the embedding model) during the meta-testing stage.

---

> > ### Comment · Reviewer_QttX · 2022-08-09
> > **Thank you for the comments.**
> >
> > After careful consideration of the rebuttal, the author's comments regarding my concerns, and the discussions pursued by other reviews, I will be maintaining my current recommendation for acceptance.

---

### Official Review · Reviewer_bRfN · 2022-07-12

**Rating:** 6
**Confidence:** 4
**Soundness:** 3 good
**Presentation:** 4 excellent
**Contribution:** 2 fair

**Summary:**

This work proposes a novel negative pseudo-labeling algorithm to tackle semi-supervised few-shot learning. The key insight is negative labels are easier to predict, therefore, pseudo labels on unlabeled samples can be better predicted by iteratively predicting negative labels until all the negative ones are excluded. Extensive experiments have been conducted on four few-shot learning benchmark and show better performance than SOTA.

**Questions:**

I would suggest the authors to address my concerns regarding the marginal improvement and hard negative classes, as listed in the weakness.

Post-rebuttal
The authors addressed all of my concerns. Therefore, I would recommend a Weak Accept.

**Limitations:**

I cannot find the limitations and potential negative societal impact in this paper.

**Strengths And Weaknesses:**

Strengths

1. The idea of generating pseudo-labels by gradually rejecting negative labels is novel and interesting.
2. The experiments are quite extensive, including results on four public benchmarks and many analysis.
3. The paper is written well and easy to follow.

Weakness

1. Although some negative labels may be indeed easier to be predicted than the positive one, there still exist hard negative classes that are equally hard to be recognized. Those hard negative classes are in fact the most important information for learning a good classifier.  I am missing how this work can handle this case.

2. Although the method is simple and novel, the achieved improvement over SOTA is marginal (less than 1%) in most of cases, see Table 1.

Post-rebuttal

My concerns about the weakness have been addressed.

---

> ### Author Response · Authors · 2022-07-28
> **Thank you for the comments. Below please find our responses to some specific comments.**
>
> *Comment_1: Although some negative labels may be indeed easier to be predicted than the positive one, there still exist hard negative classes that are equally hard to be recognized. Those hard negative classes are in fact the most important information for learning a good classifier. I am missing how this work can handle this case.*
>
> Response_1: Our strategy for dealing with hard negative classes is a series of successive exclusion operations. Specifically, after excluding easily predicted negative labels, the remaining classes are falsely annotated with a reduced probability of false pseudo-labeling. Thus, the original hard negative classes may be correctly pseudo-labeled in such a gradually negative label rejecting way. However, the reality is that all algorithms have hard (negative) classes that cannot be solved. For these unsolvable hard classes, our MUSIC adopts a conservative strategy, i.e., introducing the reject option to ensure the correctness of the pseudo-labels. On the other hand, we agree that hard negative classes are in fact the most important information for learning a good classifier. But, explicitly and effectively handling the hard negative classes is challenging, which could depend on a sophisticated method. In that case, it would destroy the simplicity and scalability of our approach.
>
> ***
> *Comment_2: Although the method is simple and novel, the achieved improvement over SOTA is marginal (less than 1%) in most of cases, see Table 1.*
>
> Response_2: As stated in Ln. 188 of the paper, all the results are the **_average_** accuracy, which is not the result of a single experiment. Therefore, even for some cases (whose improvement is less than 1%) in Table 1, it is statistically significant. To clearly show a convincing comparison in Table 1, we conduct the pairwise $t$-test at a 95% significant level on our MUSIC with the compared methods, which are presented in the following table. “$\bullet$ ($\circ$)” indicates that MUSIC is significantly better (worse) than the corresponding method.
>
> |             | _miniImageNet_|_miniImageNet_|_tieredImageNet_|_tieredImageNet_|_CIFAR-FS_|_CIFAR-FS_|_CUB_|_CUB_|
> |:-----------:|:--------------:|:--------------:|:----------------:|:----------------:|:--------------:|:--------------:|:--------------:|:----------------:|
> ||1-shot|5-shot|1-shot|5-shot|1-shot|5-shot|1-shot|5-shot|
> |MAML|48.70$\bullet$|63.11$\bullet$|51.67$\bullet$|70.30$\bullet$|58.90$\bullet$|71.50$\bullet$|54.73$\bullet$|75.75$\bullet$|
> |ProtoNet|49.42$\bullet$|68.20$\bullet$|53.31$\bullet$|72.69$\bullet$|55.50$\bullet$|72.00$\bullet$|50.46$\bullet$|76.39$\bullet$|
> |LEO|61.76$\bullet$|77.59$\bullet$|66.33$\bullet$|81.44$\bullet$|—|—|—|—|
> |CAN|63.85$\bullet$|79.44$\bullet$|69.89$\bullet$|84.23$\bullet$|—|—|—|—|
> |DeepEMD|65.91$\bullet$|82.41$\bullet$|71.16$\bullet$|86.03$\bullet$|74.58$\bullet$|86.92|75.65$\bullet$|88.69$\bullet$|
> |FEAT|66.78$\bullet$|82.05$\bullet$|70.80$\bullet$|84.79$\bullet$|—|—|73.27$\bullet$|85.77$\bullet$|
> |RENet|67.60$\bullet$|82.58$\bullet$|71.61$\bullet$|85.28$\bullet$|74.51$\bullet$|86.60$\bullet$|82.85$\bullet$|91.32$\bullet$|
> |FRN|66.45$\bullet$|82.83$\bullet$|72.06$\bullet$|86.89$\bullet$|—|—|83.55$\bullet$|92.92|
> |COSOC|69.28$\bullet$|85.16$\bullet$|73.57$\bullet$|87.57$\bullet$|—|—|—|—|
> |SetFeat|68.32$\bullet$|82.71$\bullet$|73.63$\bullet$|87.59$\bullet$|—|—|79.60$\bullet$|90.48$\bullet$|
> |MCL|69.31$\bullet$|85.11$\bullet$|73.62$\bullet$|86.29$\bullet$|—|—|85.63$\bullet$|93.18|
> |STLDeepBDC|67.83$\bullet$|85.45|73.82$\bullet$|89.00$\bullet$|—|—|84.01$\bullet$|**94.02$\circ$**|
> |TPN|52.78$\bullet$|66.42$\bullet$|55.74$\bullet$|71.01$\bullet$|—|—|—|—|
> |TransMatch|60.02$\bullet$|79.30$\bullet$|72.19$\bullet$|82.12$\bullet$|—|—|—|—|
> |LST|70.01$\bullet$|78.70$\bullet$|77.70$\bullet$|85.20$\bullet$|—|—|—|—|
> |EPNet|70.50$\bullet$|80.20$\bullet$|75.90$\bullet$|82.11$\bullet$|—|—|—|—|
> |ICI|69.66$\bullet$|80.11$\bullet$|84.01$\bullet$|89.00$\bullet$|76.51$\bullet$|84.32$\bullet$|89.58$\bullet$|92.48$\bullet$|
> |iLPC|70.99$\bullet$|81.06$\bullet$|85.04|89.63$\bullet$|78.57|85.54$\bullet$|90.11$\bullet$|—|
> |PLCM|72.06$\bullet$|83.71$\bullet$|84.78$\bullet$|90.11$\bullet$|77.62$\bullet$|86.13$\bullet$|—|—|
> |**Ours**|**74.96**|**85.99**|**85.40**|**90.79**|**78.96**|**87.25**|**90.76**|93.27|
>
> As observed, except for very few cases, our MUSIC is significantly better than other comparison methods across different datasets. Furthermore, to further test the significance of differences between these SSFSL methods, we also employ the Friedman test (at significance level $\alpha = 0.05$) and show the result as https://anonymous.4open.science/r/MUSIC-Friedman-Test/FriedmanTest.png  We can see that our MUSIC ranks at the first place and significantly outperforms other methods.

---

> > ### Comment · Reviewer_bRfN · 2022-08-05
> > **Response to the rebuttal**
> >
> > I want to thank the authors for the rebuttal and the new significance test. My major concerns have been convincingly addressed. Therefore, I would recommend a Weak Accept.

---

> > > ### Author Response · Authors · 2022-08-06
> > > **Thank you**
> > >
> > > Thank you so much for the feedback and your recommendation to accept our work.

---

### Meta-Review · Area_Chair_axeD · 2022-08-29

**Recommendation:** Accept
**Confidence:** Certain

**Metareview:**

This paper aims to improve semi-supervised few shot learning by utilizing negative pseudo-labels. The authors report significant improvement over the previous methods in this setting. The reviewers originally had concerns about the significance of the results, but after the discussion period they all supported acceptance more than they supported rejection.

Given the simplicity of the method, the size of the improvements, and the unanimous agreement from the reviewers, I support the acceptance of this paper. While the authors improved the paper significantly during the discussion stage, I would urge them to keep working on the presentation and writing for the camera-ready version. There are still writing mistakes throughout the paper, and the meaning of some of the sentences is not clear.

**Award:**

No

---

### Decision · Program_Chairs · 2022-09-14

Accept